# BRAF Mutations and Dysregulation of the MAP Kinase Pathway Associated to Sinonasal Mucosal Melanomas

**DOI:** 10.3390/jcm8101577

**Published:** 2019-10-01

**Authors:** Maria Colombino, Panagiotis Paliogiannis, Antonio Cossu, Valli De Re, Gianmaria Miolo, Gerardo Botti, Giosuè Scognamiglio, Paolo Antonio Ascierto, Davide Adriano Santeufemia, Filippo Fraggetta, Antonella Manca, Maria Cristina Sini, Milena Casula, Grazia Palomba, Marina Pisano, Valentina Doneddu, Amelia Lissia, Maria Antonietta Fedeli, Giuseppe Palmieri

**Affiliations:** 1Institute of Biomolecular Chemistry, National Research Council (CNR), Traversa La Crucca 3, 07100 Sassari, Italy; colombinom@yahoo.it (M.C.); antonella.manca@cnr.it (A.M.); mariacristina.sini@cnr.it (M.C.S.); casulam@yahoo.it (M.C.); graziap68@yahoo.it (G.P.); marina.pisano@cnr.it (M.P.); 2Department of Medical, Surgical and Experimental Sciences, University of Sassari, Viale S., 07100 Sassari, Italy; panospaliogiannis@gmail.com; 3Anatomia Patologica, Azienda Ospedaliero Universitaria (AOU), 07100 Sassari, Italy; cossu@uniss.it (A.C.); valentinadoneddu@gmail.com (V.D.); a_lissia@yahoo.it (A.L.); mariaant.fedeli@hotmail.it (M.A.F.); 4Immunopathology and Cancer Biomarkers, Centro di Riferimento Oncologico (CRO), 33081 Pordenone, Italy; vdere@cro.it; 5Department of Medical Oncology, Centro di Riferimento Oncologico (CRO), 33081 Pordenone, Italy; gmiolo@cro.it; 6Istituto Nazionale Tumori “Fondazione Pascale”, 80131 Naples, Italy; g.botti@istitutotumori.na.it (G.B.); giosco80@gmail.com (G.S.); paolo.ascierto@gmail.com (P.A.A.); 7Oncologia Medica, Ospedale Civile, 07041 Sassari, Italy; davidesanteufemia@gmail.com; 8Anatomia Patologica, Azienda Ospedaliera Cannizzaro, Via Messina 829, 95126 Catania, Italy; filippofra@hotmail.com

**Keywords:** cancer pathogenesis, sinonasal mucosal melanoma, mutation screening, BRAF gene, MAPK pathway

## Abstract

Sinonasal mucosal melanoma (SNM) is a rare and aggressive type of melanoma, and because of this, we currently have a limited understanding of its genetic and molecular constitution. The incidence among SNMs of somatic mutations in the genes involved in the main molecular pathways, which have been largely associated with cutaneous melanoma, is not yet fully understood. Through a next-generation sequencing (NGS) approach using a panel of 25 genes involved in melanoma pathogenesis customized by our group, we performed a mutation analysis in a cohort of 25 SNM patients. Results showed that pathogenic mutations were found in more than 60% of SNM cases at a somatic level, with strikingly 32% of them carrying deleterious mutations in the BRAF gene. The identified mutations mostly lack the typical UV signature associated with cutaneous melanomas and showed no significant association with any histopathological parameter. Oncogenic activation of the BRAF-depending pathway, which may induce immune tolerance into the tumour microenvironment (i.e., by increasing the VEGF production) was poorly associated with mutations in genes that have been related to diminished clinical benefit of the treatment with BRAF inhibitors. Screening for mutations in BRAF and other MAPK genes should be included in the routine diagnostic test for a better classification of SNM patients.

## 1. Introduction

Melanomas are malignant tumours that develop from melanocytes. Cutaneous melanomas are the most common type, but such a malignancy can arise in any organ containing melanocytes (e.g., eye, nose, mouth) [1,2,3]. Sinonasal mucosal melanoma (SNM) is an aggressive and rare type of cancer with a poor prognosis [4]. The most common symptoms are nasal obstruction and epistaxis, however symptomatology can develop late or be non-specific, delaying correct diagnosis and resulting in poorer prognosis [4]. It is one of the most common mucosal melanomas (41% of them) and comprises 1% of all melanomas [5,6]. Unlike to cutaneous melanomas, whose incidence is increasing, the incidence of mucosal melanoma is estimated to remain stable. In Sardinia, Italy, the incidence of melanomas is approximately four new cases per 100,000 inhabitants per year [7]. The incidence increases with age. More than 65% of patients are older than 60 years, and less than 3% are younger than 30 years [8]. The risk of local recurrence and metastasis are between 31%–85% and 25%–50%, respectively [9,10]. 

Patients with SNM have a poor prognosis, with five years survival rates of 20%–23% in patients aged 25–64 years and 19% in patients older than 65 years of age [11,12]. SNM occur in occult sites, therefore, sun radiation (a major risk factor for cutaneous melanoma) is unlikely to be implicated in mucosal melanomas. Exposure to formaldehyde was suggested to be a risk factor since some cases were reported among workers subject to industrial or professional exposure to this substance [13]. 

Primary SNM tumours present a more aggressive oncologic behaviour and a poorer prognosis than other subsets of melanomas, with a quite ineffective clinical control by current treatments [3]. Since SNM is uncommon and most cases are reported as isolated case reports with only few small series containing data collected over many years, it is difficult to establish evidence-based guidelines for clinical management. Consequently, optimal therapeutic strategies have not been extensively defined yet. Nevertheless, mutation patterns seem to be much more heterogeneous in mucosal as compared to the cutaneous melanomas [14], and genetic profiles in such a disease may further vary in different populations and geographic areas [15]. 

From the pathogenic point of view, the complete scenario of the molecular mechanisms involved in melanomagenesis is yet to be fully clarified [16]. The mitogen-activated protein kinase (MAPK) and phosphatidylinositol 3-kinase (PI3K)-Akt are two of the most important signalling pathways in melanoma with growing evidence of their involvement in both melanoma initiation and therapeutic resistance [16]. KIT is the most frequently mutated gene in melanomas arisen in mucosae worldwide (10%–30%), with an unequal geographical distribution of the mutation prevalence according to the population of the origin of the patients [17,18,19,20,21]. Among others, alterations in components of the MAPK signalling pathway were also found to participate at variable extent to pathogenesis of the different subtypes of mucosal melanoma [14,22]. Recently, a more extensive catalogue of driver mutations in sinonasal mucosal melanomas has been reported [23,24,25], either confirming the involvement of both the MAPK and the PI3K-Akt pathways or suggesting a role for additional candidate genes (NF1, SF3B1, TP53, SPRED1, ATRX HLA-A, and CHD8).

Overall, there are indeed substantial molecular differences among melanomas according to the anatomical site of onset. For example, cutaneous melanomas closely associated with ultraviolet exposure are characterized by activating mutations of BRAF and NRAS, with loss of the PTEN expression [21,26,27]. Conversely, mucosal and acral melanomas frequently harbour mutations and/or amplifications of KIT, often associated with cyclin D1 (CCND1) or cyclin-dependent kinase 4 (CDK4) amplifications, but very rarely contain BRAF mutations [16,18,28,29,30]. Differences in epidemiology and genetic mutations arising from these different sites have led to the creation of a "melanoma molecular disease model" that classifies the disease into molecular subtypes to provide information to accelerate the research and the development of tailored targeted therapies [31]. 

To increase the understanding of the prevalence and spectrum of mutations in an infrequent subtype of mucosal melanoma, we analyse next-generation sequencing (NGS) data from 25 primary sinonasal tract mucosal melanomas, and we report several somatic mutations which may be useful to make routinely applicable for the diagnostic use in SNM.

## 2. Materials and Methods

### 2.1. Study Design

This is a retrospective study recruiting 29 patients with melanoma treated at different Italian centres as part of the Italian Melanoma Intergroup (IMI) from December 2003 to December 2013. Eligible patients had a histologically proven diagnosis of SNM and had primary tumour tissue samples available for NGS molecular analysis, regardless the disease stage and the type of clinical treatment at the time of enrolment. We recruited the patients in an unbiased fashion. Four patients were excluded from the study due to the poor quality of the DNA extracted from formalin-fixed paraffin-embedded (FFPE) tissue. The diagnosis was confirmed in all cases by reviewing the histological slides and immunohistochemical markers. Using light microscopy, the neoplastic portion of each tissue section was isolated in order to obtain tumour samples with at least 80% of tumour cells. Briefly, tissue sections put on slides underwent tumour macro-dissection using a single edge razor blade and a marked hematoxylin/eosin slide as a guide in order to remove unwanted tissue parts and enrich the specimen with malignant cells.

All patients were informed about the use of their tumour tissues samples for mutation analyses, gave the permission to collect tissue specimens for such purposes, and signed a written consent. The study was approved by the Committee for the Ethics of the Research and Bioethics of the National Research Council (CNR; n. 0031325/2013). Medical records were used for collecting clinical and pathological data (clinical presentation, tumour size, treatment modalities, and follow-up). Patient and tumour characteristics are summarized in Table 1. 

### 2.2. DNA Extraction and Next Generation Sequencing

FFPE sections were subjected to DNA extraction performed with the GeneRead DNA FFPE Kit (Qiagen, Hilden, Germany) following manufacturer´s instructions. The GeneRead FFPE purification procedure includes enzymatic removal of artificial C > T mutations caused by cytosine deamination that are critical in next-generation sequencing (NGS). These artifacts are caused by formalin fixation and aging and result in sequencing errors.

Quantity of DNA extracted was assessed using Qubit 2.0 fluorometer (Invitrogen, Life technologies) and Qubit dsDNA HS (High Sensitivity) Assay Kit (Life Technologies, Carlsbad, CA, USA).

A custom panel targeting 25 genes associated with melanoma pathogenesis was designed using the Ion AmpliSeq™ Designer. The complete description of the IMI Somatic Panel has been reported by Manca et al. [32]. Briefly, it consists of 343 amplicons of max 175bp in length, organized in three pools of primers to identify variants in coding region and splice sites for intron and exon boundaries for such 25 genes (*ARID2-DDX3X-BAP1-BRAF-CCND1-CDK4-CDKN2A-ERBB4-GNA11-GNAQ-HRAS-KDR-KIT-KRAS-MAP2K1-MET-MITF-NF1-NOTCH1-NRAS-PIK3CA-PPP6C-PTEN-RB1-TP53*). An Ampliseq IMI Somatic Panel library (Life Technologies) was prepared for each sample. Each Amplicon library with barcode (Ion Xpress Barcode Adapters, Life Technologies) was prepared from a total of 30 ng template DNA (10 ng of DNA per primer pool) using the Ion AmpliSeq Library Kit 2.0 (Life Technologies) and purified with AMPure beads (Beckman Coulter, Brea, CA, USA). The libraries were diluted at a final concentration of 50 pM, pooled together, and placed into the Ion Chef for emulsion PCR and Chip (316™ v2 BC) loading steps, then subsequently sequenced on the Ion PGM using the Ion Hi-Q™ Sequencing Chemistry.

Data from the PGM runs were processed initially using the Ion Torrent platform-specific pipeline software Torrent Suite, V.5.0, to generate sequence reads, trim adapter sequences, filter, and remove poor signal-profile reads. Initial variant calling was generated using Torrent Suite Software V.5.0 using the variant caller plug-in. The genomic reference sequence used was genome GRCh 37/hg 19. In order to eliminate erroneous base calling, three filtering steps were used to generate final variant calling. The first filter was set at an average total coverage depth of >100, each variant coverage as more than three, a variant frequency of each sample >0.03. The second filtering step was employed by visually examining mutations using Integrative Genomics Viewer (IGV) software [33] as well as by filtering out possible strand-specific errors, like a mutation that was only detected in the ‘+’ or ‘−‘ DNA strand, but not in both strands. The third step was filter selecting only non-synonymous single-nucleotide variants (ns-SNVs).

For each variant, pathogenicity was assessed through data comparisons using the following sequence databases: the ClinVar archive of reports of relationships among medically relevant variants and phenotypes [34] and the Catalogue Of Somatic Mutations In Cancer (COSMIC) [35].

Pathogenic variants of the mostly-mutated and mainly-involved candidate genes (BRAF, KIT, KRAS/NRAS, and CDKN2A) on the Ion Ampliseq IMI Somatic Panel were all confirmed by sequencing gene-specific amplicons using a Sanger-based approach (primer sequences and pairs can be provided upon request). Briefly, all PCR-amplified products were directly sequenced using an automated fluorescence-cycle sequencer (ABI3130, Life Technologies). Sequencing analysis was conducted in duplicate and in both directions (forward and reverse) for all evaluated samples. Chromatograms from the Sanger sequencing assays confirming the pathogenic mutations are available on request. For the Sanger-based sequencing, PCR reactions were adjusted by increasing amplification cycles in order to improve the detection of mutations with low frequencies. Adjustments were performed for patients SN04, SN08, and SN17.

### 2.3. Statistical Analysis

Molecular and clinical data were registered in a dedicated digital database, and statistical analysis was performed. Results were expressed as percentages, means, and standard deviations (SD). Statistical significance was tested with the chi square test or Fisher’s exact test, as appropriate. Statistical tests were considered significant when the corresponding p values were <0.05. The analyses were performed using MedCalc for Windows, version 15.4 64 bit (MedCalc Software, Ostend, Belgium).

## 3. Results

### 3.1. Clinical Features

A total of 29 patients with ascertained diagnosis of sinonasal mucosal melanoma were identified through a multicentre retrospective study. Four were excluded because of the poor quality of the DNA extracted. Melanoma arises in the nasal cavity in 13 cases (52%) and in the paranasal sinuses in 12 (48%) cases (Table 1). 

The average age of the patients was 70 years, ranging from 49–91 years, of which 16 were men (62%). The median follow-up time was 36 months (range 9–78 months). 17 patients died due to the disease and five died due to other causes. Currently, two patients are living with the disease, and another without the disease (Table 1). 

Overall, 36% of the analysed samples were from Sardinian patients (9/25, 36%). Sardinia is a geographically isolated region with a high inbreeding rate and a homogeneous genetic background of the population. The remaining 64% of cases (16/25, 64%) were from different areas in Southern Italy, all characterized by high levels of genetic heterogeneity (comparable to those of the general populations).

### 3.2. Mutation Analysis

We analysed the coding regions of a designed custom panel of 25 known genes associated with cell cycle dysregulation, differentiation, angiogenesis, and oncogenesis by NGS at somatic level only. The multi-gene panel called IMI Somatic panel was firstly reported by our group [32] and includes oncogenes or tumour suppressor genes underlying melanomagenesis, which were reported to be most frequently mutated in the NGS-based studies (vast majority of investigated cases were cutaneous melanomas) [14,16,22,36]. We detected a total of 118 different non-synonymous single-nucleotide variants (ns-SNVs) and seven complex mutations (non-frameshift deletion/INDEL and multiple nucleotide variant/MNV) occurring in 19 different genes in the 25 SNM tumour samples tested. For 11 samples, more than one ns-SNV was detected in the same gene like indicated by a number in the corresponding square into the Figure 1. Overall, 21/25 (84%) tissue samples were found to carry at least one ns-SNVs/gene (*n* = 80) and 20/25 (88%) harboured at least a nucleotide polymorphism among the four genes, KDR, CCND1, CDKN2A, and TP53 (total polymorphisms detected: 27) (Figure 1). Appendix A reports the full list of variants (*n* = 125) found in each tissue sample. Nearly all of them were missense mutations (*n* = 110, 88%), whereas the remaining were nonsense mutations (*n* = 8, 6.4%), a non-frameshift deletion in the CCND1 gene (*n* = 6, 4.8%) and a multiple nucleotide variant in the PIK3CA gene (0.8%) in a single case. All variants detected into the present study have been previously reported in the publicly available databases of human gene mutations (see Methods). 

Figure 1 shows aggregate interpretation of variant for each individual patient. Eight patients (32%) harboured a mutated BRAF gene. The pathogenic exon 15 V600E variant constitutively activating BRAF was detected in seven (28%) cases, whereas variant resulting in the BRAF lack function (e.g., D594E) occurred in the remaining one case (0.7%). We also identified pathogenic mutations in the KIT, RAS (1 KRAS, 2 NRAS), CDKN2A, ARID2, CDK4, NF1, CCND1, PTEN, and PP6C genes in 12 patients (48%), in four cases, with coexistence of *BRAF* mutations (Figure 1, Appendix A). Concurrence of pathogenic mutations in melanoma driver genes was observed in a fraction of cases from our series (six out of 25 patients, 24%, Figure 1), while single nucleotide polymorphisms in KDR, CCND1, CDKN2A, and TP53 genes, and non-synonymous variances of unknown functional significance (VUS) largely occurred in the tissue samples (23/25, 92%) (Figure 1). The spectrum of individual-gene variants found in each tissue sample is graphically illustrated in Figure 2.

The major types of DNA damage induced by ultraviolet radiation are provided along with the principle mechanism of repair and common mutagenic effects. In our series, double-strand single-nucleotide mutations were found equally distributed between transitions (C > T or A > G, 53%) and transversions (A > C-T or G > C-T, 47%, Figure 3). We further verified the frequency of the nucleotide substitution and the typical UV signature (C > T) in SNM cases from our series. A total of 28% of C > T mutations occurred in our series, whereas this nucleotide change is mostly predominant in cutaneous melanomas (up to 90%) [22]. As expected, this is further confirmation that the UV exposure is not involved into the SNM pathogenesis.

Finally, patients with high mitotic rate (11/25, 44%), a prognostic factor that reflects the biological behaviour of the tumour, showed pathogenic gene mutations in vast majority of cases (9/11, 81.8%), though association was not significant (Table 2). Analogously, the presence of pathogenic mutations was independent of all remaining personal and histopathological parameters including tumour ulceration and necrosis giving rise to bleeding (Table 2).

From the practical point of view, the herein proposed the selected panel of genes by NGS was demonstrated to be a useful tool for the classification of SNM variant subtypes, even starting from small amount of paraffin-embedded archival tissues. This may represent an added value in the perspective to transfer the mutation analysis as a diagnostic tool in clinical practice.

## 4. Discussion

Genomic mutations are present in most melanomas and the identification and association with cutaneous melanomas has been extensively investigated [22]. SNM is a highly specific rare form of mucosal melanoma, often diagnosed late and with poor prognosis [4]. Therefore, genetic studies of this tumour type have been majorly limited by sample sizes with negative impact of advancement in understanding of the prognostic factors and a delayed start design for optimal SNM treatment [37,38]. 

In the present study, we performed the characterization of a cohort of 25 SNM patients, including NGS analysis of 25 genes frequently mutated in melanomas. This represents an opportunity to identify somatic mutations specifically associated with the SNM subtype useful in identification, classification, and prognosis of the disease. Genetic testing showed that pathogenic mutations occur frequently in SNM tissue samples (16/25, 64%), with 32% of these carrying a mutated BRAF gene. Concurrence of pathogenic mutations was observed in 24% of cases. According to literature data, some mutations frequently found in cutaneous melanoma, such as pathogenic variants of PTEN, TP53, or RB1 [14], were rare in our SNM series, while pathogenic mutations in KIT was reported to be most common in mucosal melanoma as well as in our series (12%) [7,18,30]. We also identified numerous non-synonymous variants of unknown functional significance, which in the future might be clarified for their genetic biological effects and their role on pathological aspects in the SNM subtype. 

Overall, results here obtained strongly support the pathogenic role of driver mutations into the RAS/BRAF/MAPK pathway in SNM. Strikingly, we observed frequent oncogenic variations in BRAF (32% of cases) and in addition of RAS (1 KRAS, 2 NRAS, 12%) and KIT (12%) in our series. Therefore, approximately three fifths of SMN patients presented a mutated driver oncogene that lead to activation of the MAPK pathway. The association of a high frequency of BRAF mutations with SNM is somehow surprisingly, as previous studies, argued against it [36,37,38]. Exon 15 BRAF V600E variant was most frequent in cutaneous melanoma with a UV signature [39,40,41]. Recently, some studies reported the prevalence of BRAF mutations as lower than 10% in SNM patients [24,42,43]. In the present study, data revealed that SNM lack the mutational signature of UV light, predominantly of transversion mutation, in overall genes tested, thus unveiling a different mutational signature between cutaneous and SNM subtypes. Although one could argue that such a difference might somehow be due to the use of our DNA extraction kit which removes C > T artefacts formed during tissue sample preparation (see Methods), the UV signature has been largely demonstrated not to be affected or diminished at all in other studies by our group using the same standardized kit protocol [7,27,32]. Moreover, our findings are consistent with those reported by Newell and colleagues, strongly indicating a low ultraviolet radiation-related mutation load in sinonasal and oral mucosal melanomas [24]. Therefore, data are in line with the hypothesis that continuous sun exposure is not a significant risk factor in SNM [20,26,37]. 

Several studies have reported that melanoma cells carrying active BRAF mutations are able to increase secretion of immunosuppressive cytokines, lower expression of tumour-associated antigens, and upregulate checkpoints ligands as PD-L1/L2, leading to the suppression of T-cell activities [44,45,46]. As a confirmation of this, an increased T cell infiltration and upregulation of melanoma antigens in patients treated with BRAF inhibitors has been demonstrated [46,47]. Moreover, BRAF mutations may also promote immunosuppressive effects into the tumour microenvironment (TME) by inducing the production of VEGF and promoting the CD73 expression and subsequent increased levels of adenosine [47,48,49]. Overall, these events strongly contribute in establishing an immune tolerance in TME by favouring the accumulation of immunosuppressive cells, such as myeloid-derived suppressor cells MDSCs and regulatory T-cells (Tregs), and/or by inhibiting the accrual of activated T lymphocytes and the maturation of dendritic cells [49,50]. Figure 4 summarizes the effects of the oncogenic activation of the MAPK pathway on TME, and emphases the role of KIT, RAS, and BRAF alterations in the activation of this pathway.

Finally, as recently underlined by Wong and colleagues [25], the occurrence of several activating mutations in the MAPK pathway (NRAS, BRAF, NF1, and KRAS) in mucosal melanoma and SNM open the way to select patients with tumours carrying actionable driver mutations which could be targeted using specific drug inhibitors.

## 5. Conclusions

The present study, carried out within the Italian population, aims to further improve our understanding of the genetic characterization of SNM at somatic level and could contribute to help the guidance of SNM patients’ management. We propose a restricted panel for BRAF, KRAS/NRAS, KIT, CCDN1, and CDK4 mutations in the routine diagnostic test in order to better classify the SNM subtypes of melanoma. In the light of the results of our study, we retain that the assessment of the BRAF mutation status should be mandatory in SNM cases due to selective inhibition of the MAPK pathway with BRAF and MEK inhibitors for unresectable, advanced metastatic melanomas [51].

In this sense, NGS is indeed a powerful tool for a more accurate identification of all main somatic mutations among SNM patients, allowing to greatly improve the efficiency and timeliness of the melanoma classification with an effective impact on the selection of the ones who may really benefit of such therapeutic options.

## Figures and Tables

**Figure 1 jcm-08-01577-f001:**
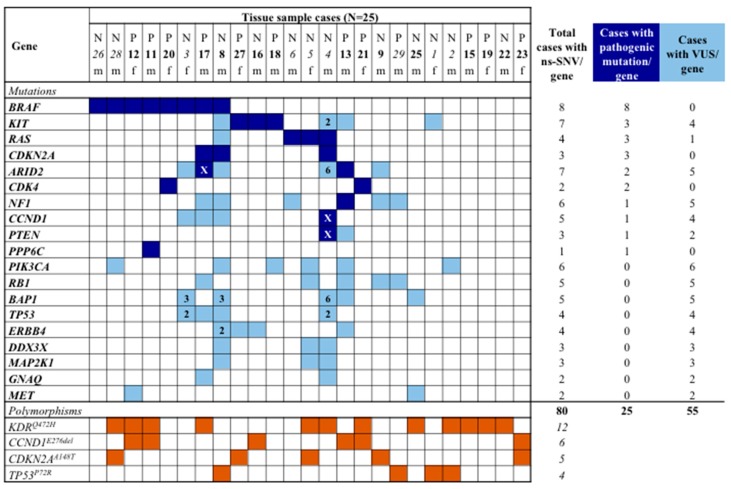
Spectrum of non-synonymous single-nucleotide variations (ns-SNV) and polymorphic variants. The table summarises the distribution of the identified mutations per gene and patient. In blue, pathogenic mutations; in light blue, variant of unknown significance (VUS); in brown, single-nucleotide polymorphisms (present in >1% general population). Number in the square indicates the total amount of VUSs in that sample. The “x” symbol in blue squares indicates the coexistence of a pathogenic mutation and a VUS into the same sample. N, nasal cavity; P, paranasal sinuses; f, female; m, male. In italic, Sardinian patients; in bold, non-Sardinian patients.

**Figure 2 jcm-08-01577-f002:**
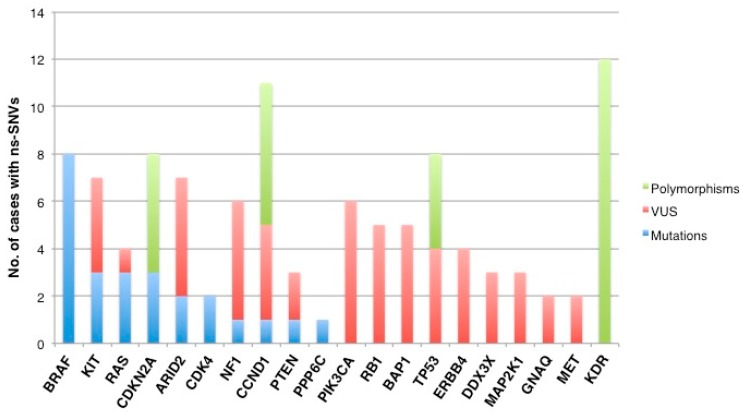
Type of non-synonymous sequence variants per gene. The graph summarises the number of pathogenic mutations (light blue bars), variants of unknown significance (red bars), and single-nucleotide polymorphisms (green bars) found in each individual gene analysed. ns-SNV, non-synonymous single nucleotide variants. VUS, variant of unknown significance. Mutations are intended as pathogenic variants.

**Figure 3 jcm-08-01577-f003:**
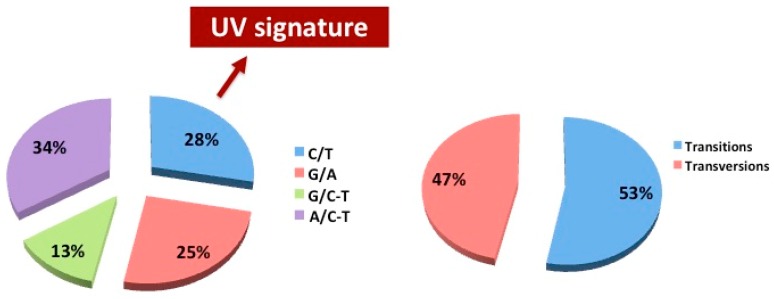
Mutation spectra in the analysed samples. (**left**) Pie chart shows the percentage distribution of nucleotide variations in the analysed genes. (**right**) Pie chart shows the percentage of transitions versus transversions mutations in the analysed genes.

**Figure 4 jcm-08-01577-f004:**
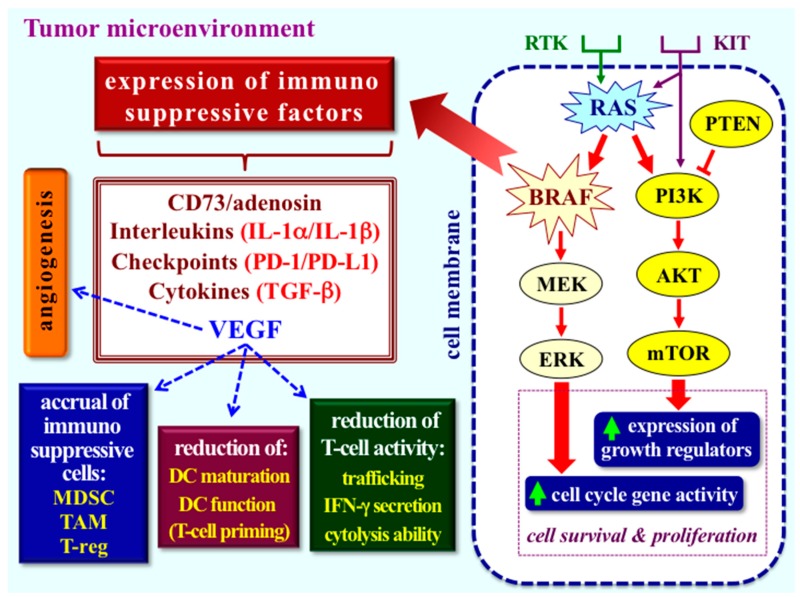
Intracellular molecular alterations and their effects on tumour microenvironment immune activity. Activated MAPK pathway in melanoma cells triggering immune escape mechanisms. Arrow, activating regulation, interrupted line, negative regulation. RTK, receptor tyrosine kinase. IFN, interferon. IL, interleukin. MDSC, myeloid-derived suppressor cell. TAM, tumour-associated macrophage. T-reg, regulatory T cell. DC, dendritic cell.

**Table 1 jcm-08-01577-t001:** Clinical and pathological characteristics of patients.

CHARACTERISTIC	Patients (*n* = 25)	%
Median age at diagnosis (range), years	70 (49–91)	
Male/Female sex	16/9	64/36
Mean follow up (range), months	36 (9–78)	
Anatomical site		
Nasal cavity	13	52
Paranasal sinuses	12	48
Histological variables		
Mitosis (<1 / >1)	14/11	56/44
Necrosis (present/absent)	10/15	40/60
Ulceration (present/absent)	14/11	56/44
Status at end of follow-up		
Death from disease	17	68
Death from uncertain cause	5	20
Alive with disease	2	8
Alive with no evidence of disease	1	4
Disease’s stage at diagnosis		
pT2	3	12
pT3	9	36
pT4	8	32
N0/N+	21/4	84/16
M0/M1	24/1	96/4
Unknown	5	20

**Table 2 jcm-08-01577-t002:** Distribution of pathogenetic mutations in the SNM cohort.

CHARACTERISTIC	Cases with Pathogenetic Gene Mutations			Cases with Mutations in BRAF + RAS Genes		
	No.	%	*p*	No.	%	***p***
Total cases (*n* = 25)	16	64.0		11	44.0	
Sex						
Female (*n* = 9)	6	66.7	1.000	4	44.4	1.000
Male (*n* = 16)	10	62.5		7	43.8	
Population origin						
Sardinian (*n* = 9)	6	66.7	1.000	6	66.7	0.115
Non-Sardinian (*n* = 16)	10	62.5		5	31.3	
Anatomical site						
Nasal cavity (*n* = 13)	8	61.5	0.881	7	53.8	0.453
Paranasal sinuses (*n* = 12)	8	66.7		4	33.3	
Mitosis						
< 1 (*n* = 14)	7	50.0	0.208	5	35.7	0.592
≥1 (*n* = 11)	9	81.8		6	54.5	
Necrosis						
Present (*n* = 10)	6	60.0	1.000	4	40.0	1.000
Absent (*n* = 15)	10	66.7		7	46.7	
Ulceration						
Present (*n* = 14)	9	64.3	1.000	6	42.9	0.783
Absent (*n* = 11)	7	63.6		5	45.5	
Disease stage at diagnosis						
pT_2-3_N_0_ (*n* = 8)	5	62.5	0.970	3	37.5	0.875
pT_4_N_0_ (*n* = 7)	5	71.4		4	57.1	
pT_any_N+/M1 (*n* = 5)	3	60.0		2	40.0	
Unknown (*n* = 5)	3	60.0		2	40.0	

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
