# Peer review of "BRAF Mutations and Dysregulation of the MAP Kinase Pathway Associated to Sinonasal Mucosal Melanomas"

_jcm, 2019, doi:10.3390/jcm8101577_

Round 1

Reviewer 1 Report

Authors of manuscript No. jcm-591090 present results of the next generation sequencing of panel of 25 selected oncogenes and tumor suppressors in quite rare cancer entity - sinonasal mucosal melanoma (SNM). Their results suggest that mutation spectra of cutaneous melanoma and SNM notably differ and that SNM share genetic profile of tumors with dysregulated MAPK signaling pathway, namely due to the high content of pathogenic BRAF gene mutations. These findings may lead to improved diagnostics of SNM and to a formulation of new therapeutic targets for this neoplasm.
The study brings interesting results in the view of molecular classification of solid tumors. Paper flows well, but presentation of results needs certain improvements for the sake of clarity and robustness of study interpretation. 

Major points
1/ Authors analyzed only tumor tissues and the study lacks controls for comparison and dissection of somatic variations from germline background. From the description of data processing it is not entirely clear if some kind of filtering through genetic data of general population to remove germline events from somatic ones was performed. Sequencing of tumor-control (tissue or blood) DNA would be optimal for this purpose, but in the case such samples are not available, at least germline variants of general Italian population should be filtered out and somatic events discussed in the paper separately from germline ones. This is particularly case of Fig. 3 and text reporting 25,313 nsSNVs observed which are presented out of context and reader does not much understand the meaning of these data – is it mix of somatic and germline variants and if yes what is its sense? Majority of polymorphisms is considered benign unless some kind of in silico functional analysis (SIFT, PolyPhen, Regulome, etc.) is performed.

2/ Description of sample processing is not clear enough. Authors state that some kind of macrodissection to get 80% of tumor cell content was performed, but how exactly was this done – by needle or other way of dissection (under microscope) or any other technique? It is also not provided at what stage of diagnostic and therapeutic process patient samples were obtained. Were all samples from incident patients, i.e. before any treatment or were there some samples from relapsed patients after adjuvant treatment?

3/ Certain results are not presented in comprehensive manner, e.g., in Table 1 all categories should be provided (males, females; N-, N+; M0, M1, etc.). Are there significant differences in mutation frequencies between patients divided by personal and clinical characteristics (Table 2)? Fig.2 may be deleted as it contains basically the same information as Fig.1 (number and type of mutations). Fig.1 should contain also information about mitosis which seems to differ by pathogenic mutation counts (Tab.2).

4/ Authors observed a highly different rate of UV signature events in SNMs (Fig.4) compared to the previously reported rate for cutaneous melanoma. I wonder whether this difference could be (at least partly) explained by the use of GeneRead FFPE DNA extraction kit which removes C>T artefacts formed during FFPE storage in the present study. Could authors discuss this eventuality in the Discussion section?

5/ Authors suggest restricted panel for diagnosis of SNMs. Why some frequently mutated genes, e.g. CDKN2A and ARID2 are not included in this panel?

Minor comments:

Fig.1 contains most probably typo in all dark blue squares with white number one which should perhaps be two.

Chapter 3.2.: at some point authors state that they found 98 non-synonymous SNVs, but few lines further they mention 110 missense mutations. Nomenclature should be unified throughout the paper.

Discussion, lines 273-285: I do not understand the formulation about BRAF mutations and UV signature in SNMs. How could data in the present study reporting 32% of BRAF mutations support hypothesis about the lack of UV signature based on data of others reporting 8% of BRAF mutations in SNMs? Authors should better explain their rationale.

Author Response

Manuscript ID jcm-591090

Please find enclosed the revised version of our manuscript. Revision of the manuscript is based on the remarks of the reviewers; the changes made are highlighted in yellow into the manuscript and here listed on a point-by-point basis.

REVIEWER 1

Major points

POINT 1: Authors analyzed only tumor tissues and the study lacks controls for comparison and dissection of somatic variations from germline background. From the description of data processing it is not entirely clear if some kind of filtering through genetic data of general population to remove germline events from somatic ones was performed. Sequencing of tumor-control (tissue or blood) DNA would be optimal for this purpose, but in the case such samples are not available, at least germline variants of general Italian population should be filtered out and somatic events discussed in the paper separately from germline ones. This is particularly case of Fig. 3 and text reporting 25,313 nsSNVs observed which are presented out of context and reader does not much understand the meaning of these data – is it mix of somatic and germline variants and if yes what is its sense? Majority of polymorphisms is considered benign unless some kind of in silico functional analysis (SIFT, PolyPhen, Regulome, etc.) is performed.

REPLY: The Reviewer correctly inferred that in our study we did not collect blood samples for mutation screening on germline DNA. On the other hand, it is not possible to filter out germline variants of general Italian population since our SNM patients originating from genetically different geographical areas: some from genetically homogeneous population in Sardinia, some from a population with highly genetic heterogeneity across the remaining parts of Italy, and some from a population with an intermediate level of genetic homogeneity in Sicily. Since the main purpose of this study was to evaluate the spectrum of pathogenetic variants at somatic level in Italian SNM cases (this is the reason to accurately process the tissue samples in order to obtain a high level of enrichment of neoplastic cells - at least 80%, as better specified into the “Methods” section as well as in the answer to Point 2, here below), we agree with the Reviewer that we can not make any speculation about the distribution of genetic variants in our series.

Therefore, we decided to completely delete the following paragraph into the “Results” section:

“Considering the high number of ns-SNVs and polymorphisms present in each individual SNM case analysed, the distribution of genetic variants resulted highly heterogeneous in our populations. Indeed, we observed a total of 25,313 non-synonymous SNVs with a distribution varying between 240 and 3,440 variants per patient”. As consequence, Figure 3 was eliminated and the remaining ones were renumbered (in total, there are four figures into the present version of the manuscript).

Finally, it has been repeatedly stated throughout the text that mutation analyses were conducted at somatic level only.

POINT 2: Description of sample processing is not clear enough. Authors state that some kind of macrodissection to get 80% of tumor cell content was performed, but how exactly was this done – by needle or other way of dissection (under microscope) or any other technique?

REPLY: Into the “Methods” section we have added the following sentence aimed at clarifying this part of the tissue sample processing: “Briefly, tissue sections - put on slides - underwent tumor macro-dissection using a single edge razor blade and a marked hematoxylin/eosin slide as a guide, in order to remove unwanted tissue parts and enrich the specimen with malignant cells”

POINT 2: It is also not provided at what stage of diagnostic and therapeutic process patient samples were obtained. Were all samples from incident patients, i.e. before any treatment or were there some samples from relapsed patients after adjuvant treatment?

REPLY: The second sentence in the “Methods” section has been modified as follows in order to specify this issue: “Eligible patients had a histologically proven diagnosis of SNM and had primary tumour tissue samples available for NGS molecular analysis, regardless the disease stage and the type of clinical treatment at the time of enrolment.”

POINT 3: Certain results are not presented in comprehensive manner, e.g., in Table 1 all categories should be provided (males, females; N-, N+; M0, M1, etc.).

REPLY: As rightly suggested by Reviewer, all categories (for each of them, numbers and percentages) have been provided in Table 1.

POINT 3: Are there significant differences in mutation frequencies between patients divided by personal and clinical characteristics (Table 2)?

REPLY: We have substitute the old Table 2 with a new one in which the “p” values are reported for each single parameter. Into the “Results” section, it has been also revised the text - since the previous one was not properly correct and thus misleading (due to the absence of any significant association), by specifying that: “patients with high mitotic rate (11/25, 44%), a prognostic factor that reflects the biological behaviour of the tumour, showed pathogenic gene mutations in vast majority of cases (9/11, 81.8%), though association was not significant (Table 2). Analogously, the presence of pathogenic mutations was independent of all remaining personal and histopathological parameters”. Accordingly, we have also modified the text within the Abstract regarding the association between mitosis and mutations. Finally, we have added a paragraph describing the statistical methodology used into the “Methods section”.

POINT 3: Fig.2 may be deleted as it contains basically the same information as Fig.1 (number and type of mutations).

REPLY: All authors retain that Figure 2 may help readers to easily visualize the distribution of the different types of variants identified (though we know that data are already contained in Figure 1).

POINT 3: Fig.1 should contain also information about mitosis which seems to differ by pathogenic mutation counts (Tab.2).

REPLY: According to the changes indicated above regarding the lack of any significance of the association between mitosis and mutations, indications of mitosis were not added into the Figure 1.

POINT 4: Authors observed a highly different rate of UV signature events in SNMs (Fig.4) compared to the previously reported rate for cutaneous melanoma. I wonder whether this difference could be (at least partly) explained by the use of GeneRead FFPE DNA extraction kit which removes C>T artefacts formed during FFPE storage in the present study. Could authors discuss this eventuality in the Discussion section?

REPLY: Into the “Discussion” section, as rightly suggested by Reviewer, it has been added the following sentence about this issue: “Although one could argue that such a difference might be somehow due to the use of our DNA extraction kit which removes C>T artefacts formed during tissue sample preparation (see Methods), the UV signature has been largely demonstrated not to be affected or diminished at all in other studies by our group using the same standardized kit protocol (7,24,29)”.

POINT 5: Authors suggest restricted panel for diagnosis of SNMs. Why some frequently mutated genes, e.g. CDKN2A and ARID2 are not included in this panel?

REPLY: The two genes indicated by the Reviewer were indeed analysed. Anyway, in addition to providing the reference of our IMI Somatic NGS panel (Manca et al; ref. no. 29), we preferred to clearly report the entire list of the 25 genes (ARID2-DDX3X-BAP1-BRAF-CCND1-CDK4-CDKN2A-ERBB4-GNA11-GNAQ-HRAS-KDR-KIT-KRAS-MAP2K1-MET-MITF-NF1-NOTCH1-NRAS-PIK3CA-PPP6C-PTEN-RB1-TP53) into the “Methods”, supposing that this may be helpful for readers.

Minor comments:

Fig.1 contains most probably typo in all dark blue squares with white number one which should perhaps be two.

REPLY: the Reviewer is right, since the number may be confounding for readers. It has been changed with the “x” symbol and into the Figure 1 legend it has been added the following sentence: “The “x” symbol in blue square indicates the coexistence of a pathogenic mutation and a VUS into that sample”.

Chapter 3.2. at some point authors state that they found 98 non-synonymous SNVs, but few lines further they mention 110 missense mutations. Nomenclature should be unified throughout the paper.

REPLY: The Reviewer is completely right that there was a little confusion about the numbers and the nomenclature. They have been corrected by specifying that: “We detected a total of 118 different non-synonymous single-nucleotide variants (ns-SNVs) and 7 complex mutations (non-frameshift deletion/INDEL and multiple nucleotide variant/MNV)...”. Nomenclature has been uniformed throughout the text as well as into the legend of Supplementary Table 1.

Discussion, lines 273-285: I do not understand the formulation about BRAF mutations and UV signature in SNMs. How could data in the present study reporting 32% of BRAF mutations support hypothesis about the lack of UV signature based on data of others reporting 8% of BRAF mutations in SNMs? Authors should better explain their rationale.

REPLY: In the “Discussion” section, at this point, it was more appropriately stated that the lack of the UV mutational signature is referred to overall genes tested (thus avoiding to point out to BRAF gene, which may become a confounding speculation), as rightly noticed by Reviewer.

Reviewer 2 Report

Brief summary

In this manuscript, the authors identified mutations in BRAF(32%), RAS(12%), and KIT(12%) genes that were not UV-signature mutations within a cohort of 25 patients with sinonasal mucosal melanoma (SNMM). They propose and discuss a diagnostic gene panel for BRAF, RAS, KIT, CCDN1, and CDK4 mutations for better classification of SNMM that could help to guide diagnosis and treatment for SNMM patients.

Significance

Sinonasal mucosal melanoma (SNMM) accounts for about 1% of melanomas and it is aggressive and difficult to treat. There is a need to identify new targets to improve the treatment of SNMM.

A recent work by Newell et al. published in Nature Communications (2019) described significantly mutated genes such as NRAS, BRAF, NF1, KIT, SF3B1, TP53, SPRED1, ATRX HLA-A, and CHD8by whole-exome sequencing of 45 tumors (17 nasal tumors). Additionally, they found that the ultraviolet radiation-related mutation load of SNMM and oral mucosal melanomas are low.

Moreover, Wong et al. Nature Communications (2019) revealed several actionable mutations in mucosal melanoma such as the activating mutations in the MAPK pathway (NRAS, BRAF, NF1, and KRAS) which could be targeted using drugs such as trametinib; Ablain et al. Science (2018) identified SPRED1 loss as a driver of mucosal melanoma using targeted sequencing to high coverage in 43 human mucosal melanomas (8 sinonasal mucosal melanomas); Wroblewska et al. The American Journal of Surgical Pathology (2019) sequenced 72 sinonasal mucosal melanomas and detected SF3B1(7%), NRAS (22%), KIT(22%), and BRAF (8%) mutations; and according to Amit et al. Molecular Diagnostics (2017) SNMM harboured BRAF(8%) and KIT(5%) mutations whereas NRASmutations were detected in 30% of SNMMs and mutation status did not affect survival outcomes but NRAS mutations could be targeted by MEK inhibitors.

Unfortunately, Colombino’s et al. research work has already been studied previously in more detail although there are a few studies of sinonasal mucosal melanomas in various racial/ethnic groups (African, Asian, Australian, Latino and Native-American, and European populations), which could explain the different percentages of mutated genes. Newell et al. (2019) is the largest genomic analysis to date of mucosal melanomas from China, Australia, the United States, and Europe.

Comments

Major issues:

Methodology

The authors must include in their methodological approach the matched normal samples to improve accuracy and identify potential variants using tumor and distinguish somatic variants from germline and loss of heterozygosity variants.

Graphical abstract

Regarding the Figure 5 (line 299), the authors should change the arrow that connects PTEN with PI3K, because PTEN negatively regulates the PI3K pathway and does not actively regulate the PI3K pathway as indicated in Figure 5.

Moreover, KIT is a Receptor Tyrosine Kinase (RTK), the activation of KIT either via its ligand or oncogenic mutation activates the RAS/RAF/MEK/ERK, PI3K/AKT, JAK/STAT, and RAC/PAK/JNK pathways but not the tumor suppressor PTEN. The right arrow that connects KIT and PTEN is wrong.

It is clear that Colombino and colleagues report a study where results are in line with recently published scientific articles. The recently published works (cited above) are the largest studies employing paired tumor/normal tissue analysis, which are more precise and accurate.

Minor issues:

The authors should specify the mutated oncogenic forms of the RAS genes: HRAS, NRAS or KRAS (line 206, 275 and 306).

Author Response

Manuscript ID jcm-591090

Please find enclosed the revised version of our manuscript. Revision of the manuscript is based on the remarks of the reviewers; the changes made are highlighted in yellow into the manuscript and here listed on a point-by-point basis.

 REVIEWER 2

Brief summary. In this manuscript, the authors identified mutations in BRAF(32%), RAS(12%), and KIT(12%) genes that were not UV-signature mutations within a cohort of 25 patients with sinonasal mucosal melanoma (SNMM). They propose and discuss a diagnostic gene panel for BRAF, RAS, KIT, CCDN1, and CDK4 mutations for better classification of SNMM that could help to guide diagnosis and treatment for SNMM patients. Significance. Sinonasal mucosal melanoma (SNMM) accounts for about 1% of melanomas and it is aggressive and difficult to treat. There is a need to identify new targets to improve the treatment of SNMM. A recent work by Newell et al. published in Nature Communications (2019) described significantly mutated genes such as NRAS, BRAF, NF1, KIT, SF3B1, TP53, SPRED1, ATRX HLA-A, and CHD8 by whole-exome sequencing of 45 tumors (17 nasal tumors). Additionally, they found that the ultraviolet radiation-related mutation load of SNMM and oral mucosal melanomas are low. Moreover, Wong et al. Nature Communications (2019) revealed several actionable mutations in mucosal melanoma such as the activating mutations in the MAPK pathway (NRAS, BRAF, NF1, and KRAS) which could be targeted using drugs such as trametinib; Ablain et al. Science (2018) identified SPRED1 loss as a driver of mucosal melanoma using targeted sequencing to high coverage in 43 human mucosal melanomas (8 sinonasal mucosal melanomas); Wroblewska et al. The American Journal of Surgical Pathology (2019) sequenced 72 sinonasal mucosal melanomas and detected SF3B1 (7%), NRAS (22%), KIT (22%), and BRAF (8%) mutations; and according to Amit et al. Molecular Diagnostics (2017) SNMM harboured BRAF (8%) and KIT (5%) mutations whereas NRAS mutations were detected in 30% of SNMMs and mutation status did not affect survival outcomes but NRAS mutations could be targeted by MEK inhibitors. Unfortunately, Colombino’s et al. research work has already been studied previously in more detail although there are a few studies of sinonasal mucosal melanomas in various racial/ethnic groups (African, Asian, Australian, Latino and Native-American, and European populations), which could explain the different percentages of mutated genes. Newell et al. (2019) is the largest genomic analysis to date of mucosal melanomas from China, Australia, the United States, and Europe.

REPLY: We agree with the Reviewer that other previously published studies - mainly, that by Newell et al. (2019) - are indeed more detailed as compared with the present one. However, as also recognized by the same Reviewer, “there are few studies of sinonasal mucosal melanomas in various racial/ethnic groups”. This is even more relevant considering that our series of SNM patients from the Italian population are originating from genetically different geographical areas across the country: some from genetically homogeneous population in Sardinia, some from a population with highly genetic heterogeneity within the remaining parts of Italy, and some from a population with an intermediate level of genetic homogeneity in Sicily.

 Major issues

Methodology: The authors must include in their methodological approach the matched normal samples to improve accuracy and identify potential variants using tumour and distinguish somatic variants from germline and loss of heterozygosity variants.

REPLY: In this study, we did not collect blood samples for mutation screening on germline DNA. Since our main purpose was to evaluate the spectrum of pathogenetic variants at somatic level in SNM cases from the Italian population, we aimed at accurately processing the tissue samples in order to obtain a high level of enrichment of neoplastic cells (at least 80%), as better specified into the “Methods” section.

For this reason, we realized that we can not make any speculation about the distribution of genetic variants in our series. Therefore, we decided to completely delete the following paragraph into the “Results” section: “Considering the high number of ns-SNVs and polymorphisms present in each individual SNM case analysed, the distribution of genetic variants resulted highly heterogeneous in our populations. Indeed, we observed a total of 25,313 non-synonymous SNVs with a distribution varying between 240 and 3,440 variants per patient”. As consequence, Figure 3 was eliminated and the remaining ones were renumbered (in total, there are four figures into the present version of the manuscript).

Finally, it has been repeatedly stated throughout the text that mutation analyses were conducted at somatic level only.

Graphical abstract

Regarding the Figure 5 (line 299), the authors should change the arrow that connects PTEN with PI3K, because PTEN negatively regulates the PI3K pathway and does not actively regulate the PI3K pathway as indicated in Figure 5.

Moreover, KIT is a Receptor Tyrosine Kinase (RTK), the activation of KIT either via its ligand or oncogenic mutation activates the RAS/RAF/MEK/ERK, PI3K/AKT, JAK/STAT, and RAC/PAK/JNK pathways but not the tumor suppressor PTEN. The right arrow that connects KIT and PTEN is wrong.

REPLY: The arrows connecting PTEN and PI3K as well as KIT and PI3K/AKT pathway have been changed in the actual Figure 4 (after figure renumbering, as specified above), as rightly noticed. In the legend of Figure 4, it has been added the following indications: “Arrow, activating regulation; interrupted line, negative regulation”.

It is clear that Colombino and colleagues report a study where results are in line with recently published scientific articles. The recently published works (cited above) are the largest studies employing paired tumor/normal tissue analysis, which are more precise and accurate.

REPLY: See our answer above.

Minor issues:

The authors should specify the mutated oncogenic forms of the RAS genes: HRAS, NRAS or KRAS (line 206, 275 and 306).

REPLY: The RAS isoforms (KRAS and NRAS) carrying pathogenetic mutations have been specified into the text.

Reviewer 3 Report

I have two suggestions;

1) Validation of NGS by Sanger sequencing should be described in more details (How many samples ? Why not all? How many variants?), Chromatograms after Sanger sequencing should be provided, Did you observed all genotypes after Sanger sequencing?)

23) Some English errors should be corrected (line 74. Kit is the mostly...)

Author Response

Manuscript ID jcm-591090

Please find enclosed the revised version of our manuscript. Revision of the manuscript is based on the remarks of the reviewers; the changes made are highlighted in yellow into the manuscript and here listed on a point-by-point basis.

REVIEWER 3

I have two suggestions;

1) Validation of NGS by Sanger sequencing should be described in more details (How many samples? Why not all? How many variants?), Chromatograms after Sanger sequencing should be provided, Did you observed all genotypes after Sanger sequencing?)

REPLY: Into the “Methods” section, the paragraph regarding the Sanger-based sequencing analysis has been rewritten in order to better clarify what we have done and correct some our previous inaccuracy. In particular, we have specified that: “Pathogenic variants of the mostly-mutated and mainly-involved candidate genes (BRAF, KIT, KRAS/NRAS, and CDKN2A) on the Ion Ampliseq IMI Somatic Panel were all confirmed...” and stated that: “Chromatograms from the Sanger sequencing assays confirming the pathogenic mutations are available on request”.

2) Some English errors should be corrected (line 74. Kit is the mostly...)

REPLY: The text has been revised for the English language.

Round 2

Reviewer 1 Report

Authors well responded to all comments of reviewers and revised the manuscript accordingly.

Author Response

Authors well responded to all comments of reviewers and revised the manuscript accordingly.

REPLY: Many thanks to the Reviewer for approving all modifications we made. Moreover, the text has been further revised for the English language.

Reviewer 2 Report

According to the authors, "REPLY: The arrows connecting PTEN and PI3K as well as KIT and PI3K/AKT pathway have been changed in the actual Figure 4 (after figure renumbering, as specified above), as rightly noticed. In the legend of Figure 4, it has been added the following indications: “Arrow, activating regulation; interrupted line, negative regulation”.

Again, Figure 4 shows PTEN-->PI3K, and it should be indicated PTEN--PI3K, as well as KIT downstream activated pathways, correctly indicated.

The introduction and discussion sections of the manuscript should be updated including the recent publications in the sinonasal mucosal melanoma topic.

Author Response

According to the authors, "REPLY: The arrows connecting PTEN and PI3K as well as KIT and PI3K/AKT pathway have been changed in the actual Figure 4 (after figure renumbering, as specified above), as rightly noticed. In the legend of Figure 4, it has been added the following indications: “Arrow, activating regulation; interrupted line, negative regulation”. 

Again, Figure 4 shows PTEN-->PI3K, and it should be indicated PTEN--PI3K, as well as KIT downstream activated pathways, correctly indicated.

REPLY: We indeed modified the Figure 4 at this point, but probably we made confusion in uploading the correct file. We have included the modified version of the Figure 4 into the actual version of the manuscript.

 The introduction and discussion sections of the manuscript should be updated including the recent publications in the sinonasal mucosal melanoma topic. 

REPLY: We have included the recent publications indicated by the Reviewer into both sections (Introduction and Discussion), and the numbering of the reference sequence has been modified throughout the text accordingly.

Into the “Discussion section”, in revising the text for including the new references, we made the following changes:

- a sentence was added into the last paragraph at page 9 (“Recently, some studies reported the prevalence of BRAF mutations as lower than 10% in SNM patients [Newell,Amit, Wroblewska]”);

- a sentence was added into the first paragraph at page 10 (“Moreover, our findings are consistent with those reported by Newell and colleagues strongly indicating a low ultraviolet radiation-related mutation load in sinonasal and oral mucosal melanomas [24]”);

- a sentence has been deleted at the end of the first paragraph in page 10 (“A recent study, which showed that missense mutations in the BRAF gene occurred only in a small subset of patients (8%), further confirms this hypothesis”);

- a sentence was added into the last paragraph at page 10 (“Finally, as recently underlined by Wong and colleagues [25], the occurrence of several activating mutations in the MAPK pathway (NRAS, BRAF, NF1, and KRAS) in mucosal melanoma and SNM open the way to select patients with tumours carrying actionable driver mutations which could be targeted using specific drug inhibitors”).